# Relating Compassion, Spirituality, and Scandal before Unjust Suffering: An Empirical Assessment

**Lluis Oviedo** [1,*] **and Josefa Torralba** [2]

1   Theology Faculty, Antonianum University, 00185 Rome, Italy
2   Theological Institute of Murcia, 30001 Murcia, Spain; jtorralba@itmfranciscano.org
*   Correspondence: loviedo@antonianum.eu

**Abstract:** Recent studies in the field of cognitive science of religion have proposed a connection between religious beliefs, theory of mind, and prosocial behaviour. Theory of mind appears to be related to empathy and compassion, and both to a special sensitivity towards unjust suffering, which could trigger a religious crisis, as has often happened and is revealed in the "theodicy question". To test such relationships, adolescents were surveyed by an exploratory questionnaire. The collected data point to a more complex, less linear interaction, which depends more on cultural factors and reflexive elaboration than cognitive structures. In general, compassion and outrage before unjust suffering appear to be quite related; compassion is related to religious practice and even more to spiritual perception.

**Keywords:** compassion; spirituality; religion; evil scandal; religious crisis





## 1. Introduction and State of the Art

A widely assumed tendency links compassion with a sharper sensitivity before unjust suffering and evil. It is less clear how both feelings are related to a more religious or spiritual sense, but intuitively, we presume that religious believers should be more affected by the scandal than unjust pain could elicit. Several questions arise in those cases, and the relationship between compassion/empathy, religion/spiritual sensitivity, and scandal before evil probably follows different paths and is rendered in more models.

Trying to summarise the available theories, we propose a first model that connects, in a straightforward and simple way, religious beliefs, social feelings, and feelings of outrage before unjust suffering, at least within the main religious traditions arising after the Axial Age. In this case, we could expect that those scoring higher in religious or spiritual indicators will sense scandal arising from unjust suffering. The rationale is quite clear: Religion is related to theory of mind, or to the ability to grasp other people's mental states (Caldwell-Harris et al. 2011; Norenzayan et al. 2012), or even to prosocial coordination and enhancement (Bloom 2012); therefore, we could expect that religious people would be more sensitive towards others, and especially towards their sufferings or hardship. Since religion in its most evolved forms also integrates a moral concern, then religious people belonging to those traditions would react more explicitly against injustice, abuse, or inflicted pain of every kind (McCauley 2011, pp. 252–68; Norenzayan et al. 2016).

A different model can be conceived to represent the relationships between religion, social sense, and scandal before evil. This alternative model does not take for granted the connections we just hinted at and assumes a more disentangled or autonomous statute for each of these feelings and perceptions. To start with, many religious expressions assume a rather individualistic style, as in the case of pre-Axial religions, like those present in classical Greece and Rome (Bellah 2011). Still in very evolved religions, those that assume most mystical forms do usually not care for the well-being of others (Kolakowski 1982, pp. 98–117). Even today, a big fringe in the broad spectrum of what can be termed "spirituality" lacks prosocial concern and appears to be unrelated to other's suffering; it

is rather about cultivating self-uplifting experiences, and is closer to aesthetic enjoyment (Flanagan and Jupp 2016, pp. 1–22). Other problems arise from the traditional views of the deep ambiguity of perceived evil. Sensitivity towards evil could be more salient in those who hold a strong belief in divine providence and grace, but at the same time, evil often becomes a scandal, and even a reason to doubt and abandon a religious faith after witnessing great evil and suffering (Reilly 1991). In these cases, strong faith in a good and merciful God is at odds with the frequent experience of abuse, mistreatment, and hard suffering. Furthermore, a recent study did not show differences regarding moral judgment between religious and non-religious participants (Rabelo and Pilati 2019).

As a result, the relationship between religious/spiritual beliefs, compassion, and the scandal of evil is far from clear and unproblematic. This relationship is possibly much more complex, and it depends on more factors and variables. Amongst these factors, we can count the aspect of religious traditions and the ways to deal with the scandal of evil, the variables concerning the different ways to encourage concern for others, and the different ways of addressing the cognitive dissonance between having faith in a loving God and experiencing suffering or witnessing harsh and unbearable pain. That sensitivity is probably connected to cultural factors, and it depends less on innate cognitive mechanisms or spontaneous perceptions and reactions towards negativity. From that perspective, the issue appears to be less rooted in linear cognitive schemas and becomes more complex, requiring a deeper analysis and cognitive elaboration, or the elaboration of an explicative model, and so it is less linked to simple and direct cognitive schemas governing religious first-level biases.

For reasons of clarity, we use the term "religion" to designate institutional or organised expressions of transcendence and its communication, and we use the term "spirituality" to designate less formal and less tradition-dependent forms of self-transcendence. We also use the term "theodicy scandal" to refer to the negative reaction from the dissonance between a religious faith and the witness of great and unjust suffering. Of course, we are aware that these three terms—"religion", "spirituality", and "theodicy scandal"—enjoy many more definitions and distinctions in the field of the study of religion and spiritual experience (Hill et al. 2000).

The literature in this field does not help very much to find a clear orientation on this issue. A considerable number of studies has been devoted to the influence that religious beliefs and practices exert in moral behaviour; a recent systematic review listed 144 entries, and since then, many more should be added (Oviedo 2016). However, the relationship between religion/spirituality and the feelings of compassion and empathy does not necessarily entail a commitment to promoting prosocial behaviour. "Compassion", according to the Oxford Dictionary, is "a strong feeling of sympathy for people or animals who are suffering and a desire to help them." This emotion has received closer attention in some papers. For instance, Saslow et al. (2013) showed that compassion is related more to spirituality than religion, giving place to what can be termed a "compassion spirituality". A few years later, Krause and Hayward published an empirical study that showed how the religious commitment of those attending services enforces gratitude and the virtue of humility. This virtue nourishes a sense of compassion, which provides meaning in life and enhances the feeling of gratitude in a self-reinforcing chain (Krause and Hayward 2015). Newmeyer et al. (2016) studied to what extent religion and spirituality could mitigate "compassion fatigue" in trauma therapists, adding a new dimension to an already complex interaction.

We were unable to find empirical studies that took into account the scandal before unjust suffering, and how it could be related to compassion and spiritual insight. The theodicy problem has received a lot of attention in philosophy of religion, but little field or experimental research has been practised. There are some exceptions: A paper by Wilt et al. (2016) examined how different theodicies are related to mental health, and an article by Abbott (2019), based on field work after the earthquakes in Haiti in 2010, analysed how such tragic events were lived in the religious faith of the affected people.

Along the line of these results, it is necessary to proceed with closer examination and testing in order to better assess current theories in the study of religion; the implications of religion and spirituality in some other basic human features, such as compassion and moral development; and the specificity of religious experience and what needs to be disentangled or distinguished at other levels.

To better discern all this, an exploratory survey with an ad hoc questionnaire was designed to identify the three variables at stake (empathy/compassion, religious/spiritual identities, and scandal before unjust evil) and to see in what way they are related. The questionnaire was distributed to high school students in Spain and Italy (414), and the data were analysed by applying standard statistical analysis to discern the variations on that relatedness. As the analysis shows, the available data point to a greater complexity in the way religion, spirituality, empathy, and theodicy scandal interact or are perceived by those young cohorts, opening new questions and deeper research on such an intriguing interaction.

## 2. Our Study: Method and Data Collection

The first step in our research consisted of designing a questionnaire tailored for teenagers and their sensitivity that covered the three main variables under study. We built a brief questionnaire that collected data on three scales: one measuring compassion and empathy, a second focusing on scandal before unjust suffering, and a third scale about religious and spiritual perception and practice. Based on former experiences, we decided to reduce the survey to 60 items; all items offered answers on a Likert progression (from "totally agree" to "completely disagree"). The first scale was composed of 23 items, mostly inspired by available scales measuring compassion (four of them were considered). The second scale had 16 items, and since we were unable to find available scales that measure that sense of scandal, this became the most innovative part of the questionnaire; examples of items in that section are:

- There is too much unjust suffering in the world.
- The wickedness of some men and women has no limit.
- If God exists, He should not allow so much pain and injustice.
- God should immediately punish those who commit evil.
- God seems absent from the worst catastrophes that occur in the world.
- God acts in a mysterious way and we do not understand how He can correct evil.
- The evil in the world is too strong and there is no way to overcome it.

At the time this scale was built, we were not aware of the more elaborate "Views of Suffering Scale" (VOSS), which gathers 30 items and results in 7 factors (Hale-Smith et al. 2012), or the simpler "Theodicy Scale" with 9 items (Daugherty et al. 2009). We found several similarities and correspondences between our scale and the ones just mentioned.

The third scale comprised 17 items and tried to measure religious practice and spiritual perception; most items were taken as a short selection from former standardised questionnaires on those topics, which were already tested by our team. Some demographic items were added at the end of the instrument. Three ethical protocol questions introduced the entire set of items. In Appendix A we offer the entire questionnaire.

The questionnaire was built as a Google Page, and offered as an application for smartphones. It was distributed in the first round to a group of 20 students to test it and check language and understandability. After a few corrections, the instrument was ready and was offered to high school students between 14 and 20 years of age in Spain (270) and Italy (144). The usual ethical protocols were observed, and permissions were obtained in each case. This was clearly a convenience sample, based on personal contacts from the involved researchers. This sample of 414 cases allowed for a first exploratory test to assess the possible interactions between the selected variables.

## 3. Our Study: Data and Analysis

### 3.1. Descriptive

The average age in this sample was 16.43 years; females were 41.1% and males 53.6%, and 4.3% chose the option "I prefer not to declare". Some interesting data concerned religious identity. As Table 1 shows, at least three big religious clusters were well represented: Catholics (as could be expected from the two nations' cultural background), no religious affiliation, and Muslims:

**Table 1.** Religious affiliations.

| Religion | Average |
|----------|---------|
| Christian Catholic | 59.7% |
| Other Christians | 0.2 |
| Muslims | 11.4 |
| Other religions | 2.2 |
| No religion | 16.9 |
| No answer | 8.7 |

Other important data concerned religious indicators. Concerning attendance of religious services, the average was M = 2.53 in a ranking from 1 ("I never attend religious services") to 5 ("I attend very often"). The item on prayer offered a similar means: M = 2.48. In any case, these data are in line with the usual—rather low—levels of religious practice among that cohort.

In the next step, a factor analysis was applied. Six relevant factors were extracted with high coefficients of reliability and the following means (see Table 2).

**Table 2.** Factor analysis, factors extracted with Varimax rotation and alpha over 0.600.

| Factor | Items | Variance | Alpha | Mean | St. Dev |
|--------|-------|----------|-------|------|---------|
| 1. Empathy | 16 | 21.59 | 0.927 | 4.14 | 0.72 |
| 2. Religiosity | 6 | 8.10 | 0.838 | 2.91 | 0.99 |
| 3. Scandal | 5 | 6.87 | 0.718 | 3.40 | 0.89 |
| 4. Spirituality | 5 | 3.89 | 0.729 | 3.40 | 0.87 |
| 5. Insensitivity | 4 | 3.13 | 0.660 | 2.13 | 0.78 |
| 6. Sense evil | 3 | 2.86 | 0.660 | 3.82 | 0.85 |

The factor analysis offered remarkably interesting insights. The first one is that the main variables appeared to be unrelatedly distributed. The first factor was called "Empathy" since most of the clustered items reflect this sensitivity, and they were inscribed in the first scale. Only three items refer to the scandal scale: "We should do much more to change a world with so much pain"; "There is too much unjust suffering in the world"; "The wickedness of some men and women has no limit". In this sense, a relatedness was perceived between empathy and the perception of evil, but not so much with most of the items referring to God or to the scandal that witnessing evil could entail, and that could have a negative impact on religious beliefs.

The second factor gathered six items on the scale of religion and spirituality. It was interesting to notice that some of the items reflect what could be termed "fuzzy religion" or spirituality; at this level, it is hard to differentiate between both dimensions, if they could ever be disentangled. Second, only one item in this factor is related to empathy: "When I am depressed and things go wrong, I remember that there are many other people in the world who feel like me" (R = 0.400). This could hint at the common idea of a link between social connectedness and spiritual feelings, as shown in another related item in the spiritual scale: "Living beings are connected in a mysterious way" (R = 0.343). In any case, spiritual views offered insights that seem to be too fuzzy.

The third factor was labelled "Scandal" and it gathered six items from the scale on scandal before unjust suffering, e.g., some of the previously mentioned items or the items "God should have made us less selfish and better" and "God should not allow psychopaths and people who like to harm". The main issue is that all of these items point to a divine responsibility, look for a theological explanation, or complain about unsatisfying available answers. However, once more, the lack of relatedness to other items in the religious or spiritual scale is striking: Scandal appeared to be quite autonomous or unrelated to religious sensitivity.

The fourth factor was labelled "Spirituality" and gathered five items, like "There are other dimensions or unknown forces that also influence our reality"; "Maybe there are other worlds different from ours, where we can live better"; "There is something in us that is immortal"; "When I contemplate the universe, I understand that there is something beyond the physical world". This last item was shared with the second factor. It is again interesting how this factor appeared to be quite detached from others, but fairly related to the second one, but in any case, relatively unrelated to compassion or scandal.

The fifth factor was labelled "Insensitivity" and gathered four items on the scale on empathy that reveal lack of concern for others: "I feel disconnected from those who tell me their problems"; "I do not feel emotionally connected to people who suffer"; "When I see someone who feels bad, I feel that I cannot relate to that person". This was the only factor in which gender plays a clear role (R = 0.51), even if it was moderately present in the first one on empathy (R = −0.244). Girls appeared to be more sensitive than boys, as was found in former surveys. Once more, this negative state appeared to be unrelated to the indicators of religiosity and scandal.

The sixth factor was termed "Sense of Evil" and gathered three items mostly on the second scale: "I cannot stand the level of evil I know"; "The evil and suffering that I know makes me uneasy and makes me think"; "Given the calamities and suffering that humans cause, we should look for help beyond humanity and its resources". However, an item from the empathy scale came out: "My heart is with those who feel unhappy" (R = 0.364). This was an expected result in this set, i.e., the idea that compassion towards others entails some sense of uneasiness regarding evil and the suffering in the world; however, it seemingly affected the religious dimension little.

### 3.2. Comparing Means and Correlations

These results invite some reflections. It is useful to compare the means for the six extracted factors and the different religious ascriptions. The questionnaire distinguished between the following clusters: Christian Catholics, other Christians, Muslims, other religions, no religious affiliation, and no answer. Looking at Table 3, it is possible to appreciate differences and convergences between the means. Upon first look, Catholics and no believers ranked very close on empathy (4.18 vs. 3.96); the same happened for the factor on scandal (3.45 vs. 3.41). A significant difference was only found in the factor for sense of evil (3.55 vs. 3.22). Then, it is interesting to compare Muslims and Catholics: Concerning empathy level, Muslims showed greater compassion (4.40 vs. 4.18), but more interestingly, Muslims appeared to be less affected by the sense of scandal (2.80 vs. 3.45). Muslims also scored similarly to Catholics for the sense of evil (3.66 vs. 3.55), whereas those belonging to other religions (just 2% of the sample) had a greater sense of scandal (3.72).

Lastly, the last outcomes indicated quite clearly that the sense of compassion appeared to be unrelated to religious beliefs for those in the same cultural background, and increased only in Muslim youngsters; the sense of scandal was disconnected from religion for the Catholic majority and 17% of unbelievers, but made a difference among Muslim students, who appeared to be less scandalised in their religious beliefs by unjust suffering. This could be the consequence of broad cultural differences rather than a difference between religious and non-religious students within the same cultural setting.

**Table 3.** Comparing means among different religious affiliations and extracted factors.

| Religious Affiliation | Empathy | Religion | Scandal | Spiritual | Insensitive | Sense of Evil |
|---|---|---|---|---|---|---|
| Catholic | 4.1841 | 3.2201 | 3.4533 | 3.5298 | 2.0253 | 3.5559 |
| Muslim | 4.4083 | 3.7239 | 2.8068 | 3.5191 | 1.9087 | 3.6652 |
| No answer | 4.0165 | 2.8486 | 3.4095 | 3.3771 | 1.9419 | 3.5886 |
| Other Christians | 4.0000 | 3.1000 | 4.5000 | 3.2000 | 2.0000 | 3.8000 |
| Other religion | 3.9219 | 2.6375 | 3.7292 | 3.4500 | 1.9714 | 3.4250 |
| Not religious | 3.9683 | 2.0309 | 3.4141 | 2.9235 | 1.9304 | 3.2235 |

The correlation table offers several interesting connections between variables (see Table 4). For instance, empathy (F1) and religiosity (F2) were quite highly correlated (R = 0.274), and still more so with spirituality (R = 0.371). Furthermore, empathy correlated highly with perceived religious scandal towards evil (0.332), but it correlated negatively with lack of sensitivity, and very strongly with perceived sense of evil. Religiosity correlated strongly with spirituality, revealing once more the difficulty in disentangling both factors; religiosity also correlated strongly with sense of evil. Religious scandal (F3) was highly correlated with the perception of evil, as was expected, and with spirituality, but not with religiosity; possibly, a factor emerged here that discriminates between both subtle dimensions of religiosity and spirituality. Spirituality (F4) was, besides what has been indicated, strongly correlated with a sense of evil, and in this case the difference with religiosity factor is more significant.

**Table 4.** Pearson bivariate correlations; * sign ≤ 0.05; ** sign ≤ 0.0001.

|  | F1 | F2 | F3 | F4 | F5 | F6 | Age | Sex |
|---|---|---|---|---|---|---|---|---|
| F1 | 1 | 0.274 (**) | 0.332 (**) | 0.371 (**) | −0.258 (**) | 0.561 (**) | −0.003 | 0.258 (**) |
| F2 | 0.274 (**) | 1 | 0.050 | 0.483 (**) | 0.060 | 0.355 (**) | −0.051 | 0.136 (**) |
| F3 | 0.332 (**) | 0.050 | 1 | 0.250 (**) | 0.136 (**) | 0.308 (**) | −0.075 | 0.016 |
| F4 | 0.371 (**) | 0.483 (**) | 0.250 (**) | 1 | 0.011 | 0.428 (**) | −0.101 (*) | 0.120 (*) |
| F5 | −0.258 (**) | 0.060 | 0.136 (**) | 0.011 | 1 | −0.117 (*) | −0.068 | −0.275 (**) |
| F6 | 0.561 (**) | 0.355 (**) | 0.308 (**) | 0.428 (**) | −0.117 (*) | 1 | 0.030 | 0.182 (**) |
| Age | −0.003 | −0.051 | −0.075 | −0.101 (*) | −0.068 | 0.030 | 1 | 0.096 |
| Sex | 0.258 (**) | 0.136 (**) | 0.016 | 0.120 (*) | −0.275 (**) | 0.182 (**) | 0.096 | 1 |

Beyond those points, it is remarkable that age did not play any role, and this is quite surprising if we consider that the sample gathered was of students between 14 and 20 years. This lack of significant correlation (except for a quite moderate negative correlation with spirituality) reveals that, at this age of adolescence, there is apparently little development regarding the analysed factors. Sex offers some other clues: As most surveys have shown, sex is correlated with empathy and perception of evil—girls are more empathic than boys—and less with religiosity and spirituality—girls are a little bit more religious. Other recent surveys were unable to distinguish between males and females for those factors.

## 4. Interpreting the Outcomes: What We Learned

This exploratory research offers a first test that allowed us to assess to what extent current theories and views on the topics under examination reflect actual feelings and beliefs. In the first place, following former surveys, the outcomes point to the autonomous character of compassion. Our sample registered a high score in this young cohort; however, it appeared to be a feeling or sentiment that belongs to one's own personality traits and is less related to other features, like religious beliefs, education, or religious practice, although some traditions could influence that specific feeling. This was the case for the Muslim group in our survey, who appeared to be more sensitive towards others. More generally, this is also an outcome that emerged after comparing means between religious and non-religious students. However, Table 3 reveals a moderate correlation between empathy and religiosity factors, which indicates that we cannot disconnect these dimensions from each other, and

that this characteristic is still more salient when spirituality is considered. This could mean that spirituality bridges or works as a mediating factor between religiosity and empathy. This connection can be read in the other direction, too: Empathy contributes to a more "spiritual" sense of religious practice.

As a second important issue, the outcomes show that to some extent, empathy/compassion is related to sensitivity towards evil and suffering, and both traits seemed to be quite related to religious scandal, or how much unjust suffering could entail doubts about divine presence, action, and goodness. However, they also appeared to be separated, which could mean that their connection depends on personal cases; we can hardly deduce a general rule from these data. Some religious people are not scandalised by unjust suffering, or vice versa—people who are not very religious could use that scandal as a reproach against religious faith. In this case, it is significant to notice that the Muslims felt less scandalised or the negative consequences of scandal. Perhaps empathy or compassion link religious faith to that scandal: In general, we can say that those who feel more empathy among the religious people are affected by scandal towards unjust suffering in greater measure.

The mental structures that could link compassion or empathy, religious or spiritual experience, and the negative feelings before evil and suffering probably follow diversified paths corresponding to distinct mental systems and strategies that are used to cope with pain and negativity. We are not keen on emphasising a universal pattern: A plurality of solutions and approaches appeared to be the norm in this rich and complex panorama. This result connects with former research on how religion might be related to theory of mind, and both to prosocial enhancement. Several studies point to a possible link between mentalising ability—which could be connected to empathy—and the religious mind (Bering 2003; Barrett 2004); however, so far, there is no evidence that supports that hypothesis in a convincing way (Reddish et al. 2016). When compared to this background, our present research suggests that empathy may play a role in religious cognition and experience, and that religion may nourish compassion as a place for expressing religious identity in a mature and self-aware way. Such a connection gives place to a religious expression among many others, one that gathers these different strands into an evolved form. In any case, that pattern would not be the only possible one. For instance, people on the autism spectrum disorder, who suffer from impairment in their empathetic capacity, are not less religious than neurotypical subjects: They just develop a different religious style or experience (Ekblad and Oviedo 2017).

As stated in our introductory notes, the collected data invite us to assume a more complex approach to religious cognition than what has been proposed when considering very elementary mental structures. Indeed, this chapter belongs rather to a more reflexive or second degree of religious cognition. This is a cognitive exercise that requires further analysis and goes deeper under the elementary cognitive structures that determine religious cognition at an initial level, e.g., when we identify supernatural agents or think about divine agency. When trying to adjust religious faith and perceived evil, injustice, and unbearable suffering, religious cognition seems to need cognitive mechanisms that are more complex than those proposed by the cognitivists of religion. When we have to cope with very negative experiences—our own or those of others—we need much better cognitive tools to make sense of such negative events. If religious faith survives despite the odds, after such cognitively discouraging perceptions take central stage, then this might be read as evidence that religious identity and beliefs have been built in a more elaborate way. Some voices have pointed out belief structures resistant to contrary evidence (Van Leeuwen 2014), and some studies have even revealed the cognitive advantages of such deceptive and impervious beliefs, which stand against the odds (Bortolotti 2020). However, once more, these studies point to a different horizon we need to explore further: how such religious beliefs are formed and how they persist despite what could be perceived as contrary evidence or major disappointments. Such an exercise belongs to a different research program, one that tries to better assess how general beliefs and religious beliefs help humans tackle and cope

with very negative perceptions and experiences without throwing us into a meaningless life (Oviedo 2019).

## 5. Discussion and Concluding Remarks

The survey we presented in this article had a limited scope: The sample was not broad nor very representative, since it gathered about 400 cases from a convenience sample of high schools in Spain and Italy. However, the collected outcomes add a little piece to the huge mosaic we are trying to compose to better understand religious dynamics and their many levels and complex interactions with related factors, like empathy, prosocial attitudes, and the cognitive challenge of evil suffering for both religious and non-religious groups.

We are confident about the value of this research. It was conducted with rigour. The instrument and the scales applied showed great reliability in our tests. Furthermore, the sample size was wide enough to extract the conclusions we arrived at.

The present study is addressed mostly to those trying to better understand the inner dynamics of religious beliefs and attitudes in a multidisciplinary way (Gonzalez-Iglesias and De La Calle 2020). We expect this study to inform about current trends in a Catholic Southern European context. For this reason, it is important for this research to be replicated in other social, religious, and cultural settings, among others, to ascertain how much cultural variance influences the interplay between religion, spiritual feelings, compassion, and scandal before suffering. These variations could add new nuances to these complex experiences and cognitive elements, and could help educators and religion scholars alike to better discern how religious beliefs react before evil and negative events.

The tools we used for our instrument could turn out to be useful for further assessments and evaluations of the interaction between religion, spirituality, empathy, compassion, and sensitivity towards unjust evil in different populations and in different age groups. Our present study is an invitation to pursue analogous research to better assess common patterns and specific differences. In this way, we might gain more insight into these intriguing issues at the heart of religious experience. We wish to contribute with the present limited research to a richer and more complex view within a scientific study of religion and move beyond models that are too reductive and unable to account for the cognitive complexity of religious experience in the varieties of human and social dimensions.

**Author Contributions:** L.O. conceptualization and design; J.T. data gathering and analysis. All authors have read and agreed to the published version of the manuscript.

**Funding:** This research received no external funding.

**Institutional Review Board Statement:** The study was conducted according to the guidelines of the Declaration of Helsinki, and approved by the Ethics Committee of Antonianum University, 2 February 2019.

**Informed Consent Statement:** Informed consent was obtained from all subjects involved in the study.

**Data Availability Statement:** To access the data, please contact the authors.

**Conflicts of Interest:** The authors declare no conflict of interest.

## Appendix A. Questionnaire on Compassion, Spirituality, and Scandal before Evil

This questionnaire was part of the project "Compassion and spirituality". The objective was to study to what extent empathy and compassion are connected to spiritual and religious sensitivity, as well as to observe the incidence of other variables, such as scandal before unjust evil in the world.

*Ethical Protocol*
- I freely consent to participate in this survey.
- I have been informed about its objectives and usefulness.
- I agree to participate with the condition that the data will be treated with the utmost confidentiality and the identity of the person who answers will never be known.

*Scale about Empathy and Compassion*

1. One of the things that makes the most sense in my life is helping other people.
2. I feel disconnected from those who tell me their problems.
3. If someone needs help, I do everything I can to help them.
4. I prefer to suffer before seeing another person suffer.
5. I feel very affected by family and friends who are in need.
6. I do not feel emotionally connected to people who suffer.
7. The world is a hostile environment that must be protected.
8. If I feel that someone is having a difficult time, I care for that person.
9. I like to be close to others in times of difficulty.
10. I realise when people are sad, even when they do not say anything.
11. When I see someone who feels bad, I feel that I can not relate to that person.
12. It makes me sad to see a person I do not know alone in a group.
13. Everyone feels bad sometimes, it is part of the human condition.
14. When a friend starts talking about their problems I try to change the topic of conversation.
15. I usually listen with patience when people tell me their problems.
16. I get very angry when I see someone who is being mistreated.
17. It is important to recognise that everyone has weaknesses and that no one is perfect.
18. Suffering is only a part of the common human experience.
19. My heart is with those who feel unhappy.
20. It makes me sad to see helpless elders.
21. Despite my differences to others, I know that sadness hits everyone equally.
22. Empathy with creation helps me feel sorry for all creatures.
23. When I am depressed and things go wrong I remember that there are many other people in the world who feel like me.

*Scale on Perception of the Scandal of Evil*

24. There is too much unjust suffering in the world.
25. The wickedness of some men and women has no limit.
26. If God exists, He should not allow so much pain and injustice.
27. God should immediately punish those who commit evil.
28. God seems absent from the worst catastrophes that occur in the world.
29. God acts in a mysterious way and we do not understand how He can correct evil.
30. The evil in the world is too strong and there is no way to overcome it.
31. The world is a place that is improving, people are becoming less bad.
32. God should have made us less selfish and better people.
33. I cannot stand the level of evil I witness.
34. God should not allow psychopaths and people who like to harm to exist.
35. The evil and suffering that I know of makes me uneasy and make me think.
36. We should do much more to change a world with so much pain.
37. I would be willing to sacrifice part of my freedom if it would help to decrease the injustice and pain in the world.
38. Given the calamities and sufferings that humans cause, we should look for help beyond humanity and its resources.
39. I am deeply saddened when I recognise my limitations and errors.

*Scale on Religious and Spiritual Sensitivity*

40. I consider myself a spiritual person, whether or not I attend religious events.
41. Sometimes I feel the presence of a mysterious force in me or in others.
42. When I contemplate the universe, I understand that there is something beyond the physical world.
43. Living beings are connected in a mysterious way.
44. I attend mass or other religious celebrations.
45. I am a person who prays or meditates.
46. I am a religious believer.
47. Sometimes I need a kind of help that cannot be provided by people or the media.
48. There are some values and ideals that I consider absolute.
49. Maybe there are other worlds different than ours, where you can live better than here.
50. There are other dimensions or unknown forces that also influence our reality.
51. The world is nothing more than what we see and know.
52. Without strong hope life would not make sense.
53. Our hope depends only on human achievements.
54. There is a mysterious force in the cosmos that guides us towards good.
55. There is something in us that is immortal.
56. In order to be happy it is important to cultivate a spiritual life.

*Demographics*

57. Sex
58. Age
59. Course
60. Religion
61. Public or private school

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
