# Peer review of "Relating Compassion, Spirituality, and Scandal before Unjust Suffering: An Empirical Assessment"

_religions, doi:10.3390/rel12110977_

Round 1

Reviewer 1 Report

The study is very interesting, but the paper needs thorough revision by a native English speaker.

Author Response

The new version of the paper has been now thoroughly edited by an English language expert and we hope the new version will be more fitting to the Journal standards.

Reviewer 2 Report

I recommend publication after incorporating several edits, which should not be too onerous and would enhance the quality and utility of the article.

1) I would replace the word "scandal" with "outrage," which resonates better in an English context. This is not a "required" recommendation, simply a suggestion to consider. Scandal evokes political and religious personal moral crises, whereas out outrage is more neutral, befitting the context.

2) A better gloss of "compassion," for instance, its etymology: to suffer with.

3) I would frame the discussion better, defining the "theodicy problem" (see Mark S. M. Scott, Pathways in Theodicy: An Introduction to the Problem of Evil) and situating the discussion more clearly in the psychology of religion, essentially as an empirical engagement with theodicy in that subfield. Perhaps read two different chapters on the psychology of religion from two Religious Studies handbooks for guidance on theoretical framing.

4) A deep edit of the English and the strange red text throughout (e.g., p. 2, line 2, it's simply "divine providence"; p. 7. line 1, it's "non-religious"; p. 8, missing space after the period before "The instrument"). The text needs rhetorical and grammatical refinement to complement the research.

5) A sense of where the research goes next: it's clearly a preliminary study that is suggestive for a larger project.

I wish the authors all the best in these and other edits.

Author Response

The new version of the paper has been deeply edited by an English expert.

Round 2

Reviewer 1 Report

The author did a good job of revising the article.

This manuscript is a resubmission of an earlier submission. The following is a list of the peer review reports and author responses from that submission.

Round 1

Reviewer 1 Report

No comments

Author Response

Nothing to say

Reviewer 2 Report

Please review sentence structure, grammar and other English language conventions.

Author Response

We have extensively revised the new version of our paper with the help of an English expert; we expect this new version would be more accurate in its style and expression.

Reviewer 3 Report

Reviewer’s Feedback

The paper has great potential in the field of cognitive science of religion and is quite timely.

However, there are areas for improvement as identified below:

-Arguments made in the first 3 paragraphs lacked appropriate supporting citations  from respected scholars in the field. For example, whose models are you drawing from? The statements made around these are quite general.

-Key definitions lacking in article(e.g.,  religion, spirituality and compassion, compassion fatigue, theodicy scandal)

-Attention needed to be given to proper citation in article (e.g., remove extra period and page number missing from direct quote; see p. 2 line 61).

-Clarity needed in writing (e.g., see lines 61-64; wording is very confusing, restructure for clarity)

-Pay attention to wordiness in writing (e.g., see lines 71-74;  make into 2 statements for better clarity)

-Specify country in Caribbean you are referring to; inference is quite general (see line 73)

-Convoluted wording; make more concise (see lines 313-317)

-Revisit reference list for consistency

-Create separate sections about limitations of methodology used. Here, you can address how rigour, validity and reliability were met with the tools implemented, etc.

-Identify audience in article and the future implications of the study to such audience. Also, consider how this article is relevant cross-culturally and/or multi/inter-disciplinary. How might thus research compare and contract to the North American, African or Caribbean context? What might be other areas for future research based on the outcome of this study? These might be potential areas to address in the article to strengthen its positionality in the field.

Hope you find the above feedback helpful.

Author Response

We are thankful to the reviewer work and suggestions to improve our paper. e have tried to address all his/her points and to reender our text more legible and accurate according to the received indications. We answer point by point to all them.

-Arguments made in the first 3 paragraphs lacked appropriate supporting citations  from respected scholars in the field. For example, whose models are you drawing from? The statements made around these are quite general.

New required quotations have been added

-Key definitions lacking in article (e.g., religion, spirituality and compassion, compassion fatigue, theodicy scandal)

Definitions added to clarify our terminology

-Attention needed to be given to proper citation in article (e.g., remove extra period and page number missing from direct quote; see p. 2 line 61).

Done. The expression “compassion spirituality” is our own and not a direct quotation

-Clarity needed in writing (e.g., see lines 61-64; wording is very confusing, restructure for clarity)

Done.

-Pay attention to wordiness in writing (e.g., see lines 71-74;  make into 2 statements for better clarity)

Done.

-Specify country in Caribbean you are referring to; inference is quite general (see line 73)

Haiti, 2010, done.

-Convoluted wording; make more concise (see lines 313-317)

Done!

-Revisit reference list for consistency

Done.

-Create separate sections about limitations of methodology used. Here, you can address how rigour, validity and reliability were met with the tools implemented, etc.

We have tried to come to terms with this suggestion.

-Identify audience in article and the future implications of the study to such audience. Also, consider how this article is relevant cross-culturally and/or multi/inter-disciplinary. How might thus research compare and contract to the North American, African or Caribbean context? What might be other areas for future research based on the outcome of this study? These might be potential areas to address in the article to strengthen its positionality in the field.

We have tried to address now these interesting points.

Round 2

Reviewer 3 Report

Hi authors,

I am pleased to see the changes you have made to the paper. They definitely helped to improve its overall quality. However, I was hoping for me work to be done on the intro section. I still find it lacking in empirical rigor and adding a few random citations really won’t suffice. To offer some guidance, I have given you some places to revisit below:

 Lines 16-17 -  odd way to start intro. Could benefit from a better overarching sentence followed by this tentative sentence. Keep in mind that the first sentence should draw your readers in and set the stage for the paper.

Lines 18-22 - You are saying a lot here. Very convoluted. Simplify sentences. For instance, a first version of what? Quite vague in wording. Also, you need citations to support these ideas. Who is this first version by? Cite author’s work who informs this model/tool/instrument, etc.

Lines 22-25 – Provide citation about this rationale. Whose work are your drawing from here?

Lines 29-30 – Again, this sentence is quite vague. For example, a different version of what? Model? Also, you need to draw from the literature here to support this model/instrument, etc.

Lines 32-34-Cite

Lines 39-40 – Cite source here.

Line 30 – “many religious expressions assume”. Here give concrete examples. Also cite authors’ work to support your stance instead of making broad generalizations.

“For reasons of clarity, we use the term “religion” as designing institutional or organized expressions; and “spirituality”, as those forms of self-transcendence less formal or less linked to traditional expressions. “Compassion” refers to an emotion that moves somebody to help others; and “theodicy scandal” is the negative reaction that affects the credibility of religious faith before the perception of great and unjust suffering.” You need to cite these definitions otherwise explicitly state that these are your definitions that you coined. However, it would be good to draw from the literature with these concepts. Also, it is odd lumping all the definitions in one place. They should be dispersed throughout the paper where they were likely first introduced for better flow of coherency of ideas.

“We are confident about the value of this research: it has been conducted with rigour: the instrument and the scales applied showed great reliability in our tests; and the sample size is wide enough to extract the conclusions we arrived at.” Grammatically incorrect with all these punctuations (see comment below). Restructure with proper punctuations. Simple and concise sentences are better and less confusing for readers. Also, revisit uses for colon and semi-colon.

In general, be mindful of using so many punctuations in one sentence. It makes it hard to follow your train of thought and impacts the quality of your writing.

Again,   these are just some areas. However, I would strongly encourage you to do another thorough read through the paper and provide appropriate citations to areas that are be strongly supported by the literature that you might have overlooked. These sources will show readers your engagement with the literature.